# Dynam3D: Dynamic Layered 3D Tokens Empower VLM for Vision-and-Language Navigation

**Zihan Wang    Seungjun Lee    Gim Hee Lee**
School of Computing, National University of Singapore
`zihan.wang@u.nus.edu, gimhee.lee@nus.edu.sg`

## Abstract

Vision-and-Language Navigation (VLN) is a core task where embodied agents leverage their spatial mobility to navigate in 3D environments toward designated destinations based on natural language instructions. Recently, video-language large models (Video-VLMs) with strong generalization capabilities and rich commonsense knowledge have shown remarkable performance when applied to VLN tasks. However, these models still encounter the following challenges when applied to real-world 3D navigation: 1) Insufficient understanding of 3D geometry and spatial semantics; 2) Limited capacity for large-scale exploration and long-term environmental memory; 3) Poor adaptability to dynamic and changing environments. To address these limitations, we propose Dynam3D, a dynamic layered 3D representation model that leverages language-aligned, generalizable, and hierarchical 3D representations as visual input to train 3D-VLM in navigation action prediction. Given posed RGB-D images, our Dynam3D projects 2D CLIP features into 3D space and constructs multi-level 3D patch-instance-zone representations for 3D geometric and semantic understanding with a dynamic and layer-wise update strategy. Our Dynam3D is capable of online encoding and localization of 3D instances, and dynamically updates them in changing environments to provide large-scale exploration and long-term memory capabilities for navigation. By leveraging large-scale 3D-language pretraining and task-specific adaptation, our Dynam3D sets new state-of-the-art performance on VLN benchmarks including R2R-CE, REVERIE-CE and NavRAG-CE under monocular settings. Furthermore, experiments for pre-exploration, lifelong memory, and real-world robot validate the effectiveness of practical deployment. The code is available at https://github.com/MrZihan/Dynam3D.

## 1   Introduction

Vision-and-language navigation (VLN) tasks [1–4] require agents to integrate three core capabilities: 1) understanding natural language instructions, 2) exploring environments and localizing targets or destinations, and 3) planning and executing navigation actions. As illustrated in Figure 1(a), recent works [5–7] have predominantly focused on using video-based large models [8–10] to develop monocular VLN systems. This is due to the practical constraint that most robots are equipped with monocular cameras instead of panoramic cameras. These models pre-trained on large-scale internet data demonstrate strong language understanding and multimodal reasoning abilities, which enable effective instruction following and continuous prediction of navigation actions toward the destination.

Despite these recent advances, several limitations still remain: 1) Video-based models struggle to capture spatial geometry and semantics in large-scale 3D environments. Our experiments reveal that this significantly hinders the ability of these models to explore extensively and correct errors effectively. 2) These models lack mechanisms for structured scene memory. This prevents the use of

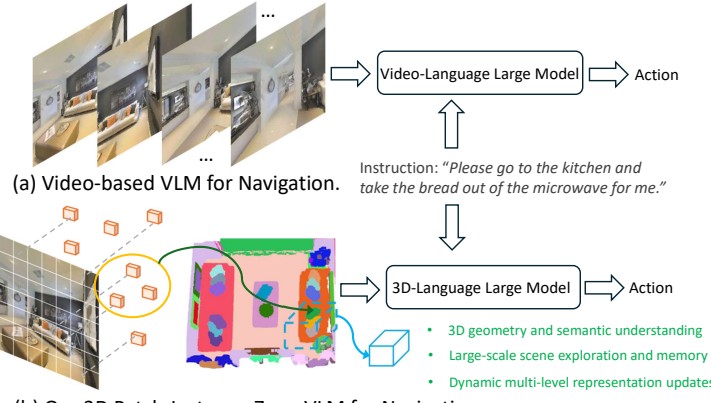

(a) Video-based VLM for Navigation.

Instruction: *"Please go to the kitchen and take the bread out of the microwave for me."*

- 3D geometry and semantic understanding
- Large-scale scene exploration and memory
- Dynamic multi-level representation updates

(b) Our 3D Patch-Instance-Zone VLM for Navigation.

Figure 1: Different vision-language large models for monocular VLN tasks. Compared to previous video-based representations (a), our Dynam3D (b) adopts dynamic hierarchical 3D representations offering advantages in spatial geometry and semantic understanding.

pre-exploration knowledge and limits the potential for lifelong learning. 3) Representations derived from historical frames are inadequate for dynamically changing 3D scenes, where frequent object and human movements lead to performance drop.

We propose Dynam3D to alleviate the limitations mentioned above. As illustrated in Figure 1(b), our Dynam3D is a 3D-language model with dynamic layered 3D representations for vision-and-language navigation. To encode 3D environments, we extract patch-level 2D features using CLIP [11] and project them into 3D space via depth maps and camera poses. Our Dynam3D employs FastSAM [12] to generate 2D instance masks, and aggregates patch features within each mask into instance-level representations. A 3D instance merging discriminator aligns 2D instances with existing 3D instances based on geometry and semantics to enable dynamic updates of 3D instance representations. Unlike previous online [13] or language-guided [14] 3D segmentation methods that focus on mask accuracy, our Dynam3D mainly aligns instance representations with the semantic space of CLIP through large-scale 3D-language pretraining to improve 3D representation quality and scene understanding.

Furthermore, our Dynam3D aggregates 3D instance features within spatial zones to facilitate understanding of large-scale environments. As a result, this enables high-level comprehension of layouts, *e.g.* bedrooms, kitchens, *etc* that instance-level features alone cannot capture. our Dynam3D updates the scene dynamically with this hierarchical patch-instance-zone representation: outdated patch features are removed when a new RGB-D observation arrives, and new features are projected and propagated across the representation layers (patch-instance-zone) for change adaptation. These features enable our Dynam3D to maintain a lifelong and dynamic environmental memory that can significantly improve navigation performance in real-world deployments.

We further introduce a generalizable feature field model [15] to render 3D patch features over an agent-centric panoramic scope for the enrichment of local geometric and semantic perception. These rendered 3D patch features combined with instance and zone representations serve as visual input to the 3D Vision-Language Model (VLM). Given language instructions and action history, the 3D-VLM directly predicts navigation actions, *e.g.*, turn $\theta$ degrees, move forward $d$ cm, or stop.

In summary, our main contributions include:

- We propose Dynam3D, a multi-level patch-instance-zone 3D representation model that performs online 3D instance and zone-level encoding and real-time hierarchical updates in dynamic environments.
- We introduce a 3D Vision-Language Model that integrates 3D patch features from generalizable feature fields and 3D instance-zone features from our Dynam3D. This balances fine-grained geometry and global spatial layout for navigation planning.
- Our monocular VLN system achieves state-of-the-art performance on benchmarks including R2R-CE, REVERIE-CE, and NavRAG-CE. The results also demonstrate our strong capabilities in pre-exploration, lifelong memory and real-world experiments.

## 2 Related Work

**Vision-and-Language Navigation.** Vision-and-Language Navigation (VLN) [1, 3, 2, 16–19] requires the agent to understand complex natural language instructions and navigate to the described destination. In contrast to early works [16–18, 20, 21] which primarily concentrate on training and evaluating models within discrete environment simulators [22, 1, 2] (*i.e.*, move on the pre-defined navigation connectivity graph, equipped with panoramic RGB-D camera), recent researches have increasingly emphasized navigation in continuous environment simulators [23–27] and the real-world deployment of monocular VLN systems [5, 28, 15, 6, 7, 29, 30]. For monocular VLN on continuous environment simulators, the agent equips only a forward-facing monocular RGB-D camera, and uses low-level actions to navigate. To leverage the language understanding and commonsense reasoning capabilities of large models, some recent works [31–35, 29] have adapted 2D-VLMs to VLN tasks, leading to notable performance improvements. Extensions such as NaVid [5], Uni-NaVid [6], and NaVILA [7] further exploit video-based large models to build high-performance monocular VLN systems with strong real-world applicability. However, video-based representations still have inherent limitations. For example, they struggle to capture fine-grained geometry semantics and comprehend large-scale spatial layouts, which in turn limits their capabilities in object localization and path planning. To the best of our knowledge, our Dynam3D is the first approach that effectively addresses the limitations inherent in previous video-based models by using a 3D-VLM to perform monocular VLN tasks in unseen and dynamic environments.

**3D Vision-Language Models.** Inspired by the development of 2D-VLM [36, 37, 8–10], recent works integrate 3D inputs, the point clouds [38–40] or multi-view images [41–43] to enable 3D scene reasoning for 3D-VLMs. These approaches differ primarily in scene representation: LL3DA [38] encodes full-scene point clouds directly; LEO [40] and Chat-Scene [39] decompose scene point clouds into object-level segments and encode corresponding features. 3D-LLM [42] and Scene-LLM [41] begin with multi-view images, apply 2D object segmentation, and aggregate CLIP features into pixel-aligned 3D points. LLaVA-3D [44] builds on a pretrained 2D VLM [37] to embed 2D patches into 3D voxels via multi-view inputs and 3D positional embeddings. This enables fast adaptation to 3D tasks while maintaining strong 2D perception. However, current 3D-VLMs face fundamental challenges in large-scale unseen and dynamic tasks such as embodied navigation. Full-scene point cloud or voxel-based representations are impractical for real-time reasoning in unseen environments. Existing models lack mechanisms for incremental updates, which make it difficult to revise or discard outdated scene information in dynamic contexts. Moreover, they struggle to balance the computational trade-off between global spatial layout and fine-grained geometric semantics. In this context, we propose Dynam3D, a 3D-VLM model that is better adapted for such dynamic embodied tasks.

## 3 Our Method

**Overview.** Figure 2 shows the framework of our Dynam3D for vision-and-language navigation. The framework takes the posed monocular RGB and depth images as input, and outputs atomic navigation actions such as turning, moving forward, stopping *etc*. Our Dynam3D maintains a set of patch feature points to encode the generalizable feature field [15] used to render the panoramic 3D patch tokens of the agent. Furthermore, our Dynam3D layer-by-layer encodes and updates 3D instance representations and large-scale cube zone representations for multi-level scene understanding and target localization (*cf.* Section 3.1). These multi-level 3D tokens, navigation instructions and history actions are then fed into a 3D-VLM for next action prediction (*cf.* Section 3.2).

### 3.1 Dynamic Layered 3D Representation Model

We first design and pre-train a multi-level 3D representation model to acquire language-aligned 3D representations encompassing both fine-grained details and global layouts.

**Encoding the Patch Feature Points.** To memorize the geometry and semantics of 3D environments, we follow HNR [45] and g3D-LF [15] in using CLIP-ViT-L/14@336px [11] as the encoder for RGB images to extract 2D patch features $\{\mathbf{g}_{t,i} \in \mathbb{R}^{768}\}_{i=1}^{I}$. $\mathbf{g}_{t,i}$ denotes the $i$-th patch feature of the 2D feature map extracted from $t$-th frame observed by the agent and $I = 24 \times 24$. The patch features $\{\mathbf{g}_{t,i}\}_{i=1}^{I}$ are then project to the corresponding 3D world coordinates $\{P_{t,i}\}_{i=1}^{I}$ using the depth map

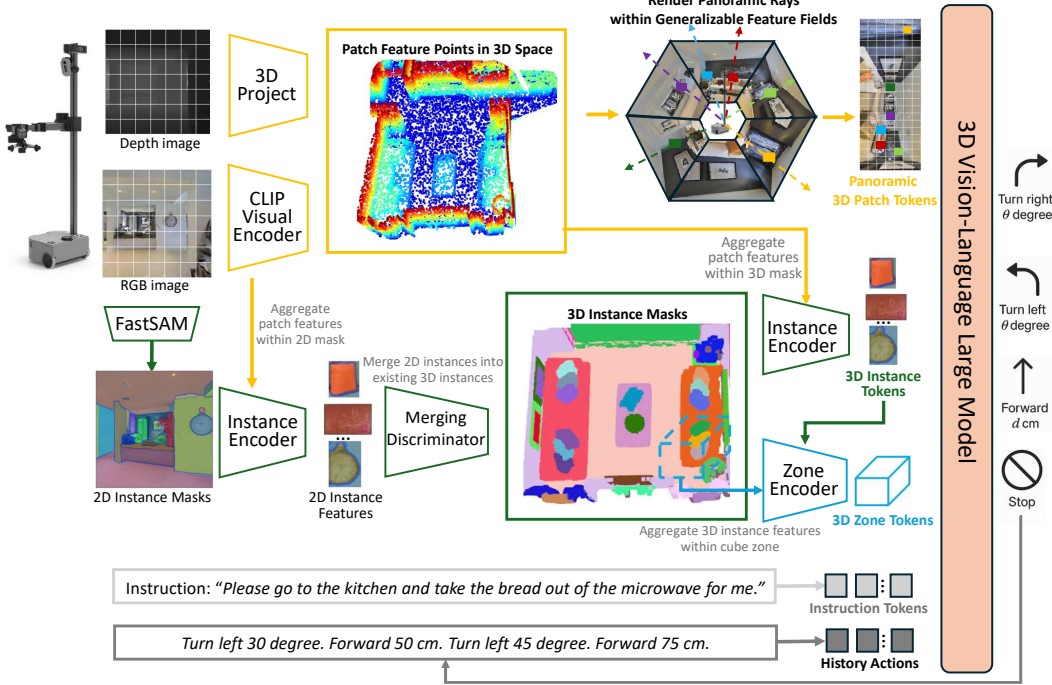

Figure 2: The architecture of our Dynam3D framework. Our Dynam3D takes posed monocular RGB and depth images as input and outputs atomic navigation actions. It encodes and updates multi-level 3D representations for scene understanding and target localization. The 3D tokens, navigation instructions and history actions are then consolidated into the 3D-VLM for next action prediction.

and camera parameters. For each feature $\mathbf{g}_{t,i}$, the observed horizontal orientation $\theta_{t,i}$ and the regional size $s_{t,j}$ are also calculated and stored to enhance the spatial representation. The set of feature points $\mathcal{M}$ can therefore be updated online as:

$$\mathcal{M}_t = \mathcal{M}_{t-1} \cup \{[\mathbf{g}_{t,i}, P_{t,i}, \theta_{t,i}, s_{t,i}]\}_{i=1}^{I}. \tag{1}$$

**Updating the Patch Feature Points.** As shown in Figure 3, we employ the *Frustum Culling* strategy to dynamically update the feature points set $\mathcal{M}$ by discarding outdated features and incorporating new ones, which differs from previous methods [46, 45, 28, 15] simply add new feature points regardless of object motion or removal. Specifically, after obtaining the observed depth image $\mathbf{D}_t \in \mathbb{R}^{H \times W}$, the frustum culling strategy transforms the 3D world coordinate $P_w \in \mathcal{M}$ of each feature point into the pixel coordinate of the depth image using the camera pose $[\mathbf{R}, \mathbf{T}]$ and camera intrinsics $\mathbf{K}$ as follows:

$$P_c^\top = \begin{bmatrix} x_c \\ y_c \\ z_c \end{bmatrix} = \mathbf{R}P_w^\top + \mathbf{T}, \quad \begin{bmatrix} u \\ v \\ 1 \end{bmatrix} = \frac{1}{z_c}\mathbf{K}\begin{bmatrix} x_c \\ y_c \\ z_c \end{bmatrix},$$

$$\text{FrustumCulling}(P_w), \text{ if } 0 < z_c < \min(d_{u,v} + \delta, \Delta), \ 0 < u < H, \text{ and } 0 < v < W. \tag{2}$$

$d_{h,w}$ denotes the depth value in row $h$ and column $w$ of the depth image $\mathbf{D}_t \in \mathbb{R}^{H \times W}$. A feature point $P_w$ is removed from the feature points set $\mathcal{M}$ by the $\text{FrustumCulling}(\cdot)$ function when $0 < z_c < \min(d_{u,v} + \delta, \Delta), 0 < u < H$ and $0 < v < W$, where $\delta$ is a noise threshold and $\Delta$ is the farthest culling distance. The frustum culling is first applied and followed by adding the new feature points when a RGB-D observation is obtained.

**Dynamically Encoding 3D Instance Representations.** Due to the overwhelming volume of 3D patch features, a direct employment as visual input to 3D-VLM is computationally and economically impractical. In contrast to voxel-level pooling approaches, *e.g.* LLaVA-3D [44], our Dynam3D encodes features at the 3D instance level since target localization in navigation instructions is mostly described in terms of object instances. As illustrated in Figure 2, FastSAM [12] rapidly segments

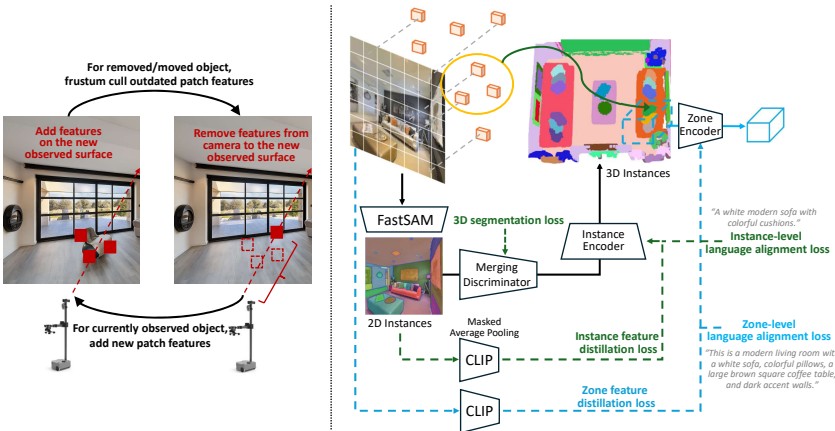

Figure 3: Left: Illustration of the feature points update and frustum culling strategy. Right: The supervision of feature distillation and 3D-language contrastive learning for our Dynam3D model.

the observed RGB image into a set of 2D instance masks. Within each mask, a transformer-based instance encoder aggregates the corresponding patch features $\{\mathbf{g}_m\}_{m=1}^M$ with positional embeddings $\{\mathbf{p}_m\}_{m=1}^M$ into a compact instance-level representation $\mathcal{O}$ using a learnable token $\mathbf{q}$ as query:

$$\mathbf{p}_m = \mathbf{MLP}(\ [P_m - \text{Average}(\{P_m\}_{m=1}^M), s_m, \cos(\theta_m), \sin(\theta_m)]\ ),$$
$$\mathcal{O} = \text{InstanceEncoder}(\mathbf{q}, \{\mathbf{g}_m \oplus \mathbf{p}_m\}_{m=1}^M). \tag{3}$$

In contrast to simple 2D instance representations, 3D instances require both multi-view and geometric consistency, enabling the agent to identify the same instance across different views. To this end, we train a Merging Discriminator to integrate 2D instance representations into consistent 3D instances, as shown in Figure 2. Initially, each 2D instance is treated as a new 3D instance. At each subsequent step, for every new 2D instance, the Top-K nearest existing 3D instances are retrieved. The Merging Discriminator evaluates each 2D–3D instance candidate pair using semantic and geometric encodings to determine correspondence. If no match is found among the Top-K candidates, a new 3D instance is created. Otherwise, the 2D instance is merged with the most similar 3D instance by concatenating their patch features and updating the 3D instance representation through the instance encoder. The 3D representation is updated with the remaining relevant patches when the outdated patches are removed via Frustum Culling. We discard the 3D instance in the case where all patches are removed.

We train the Merging Discriminator using over 5K rooms with 3D instance segmentation data: ScanNet [47], HM3D [48], Matterport3D [22] and 3RScan [49], where the annotation of instances of point clouds are processed for each point with world coordinate and instance ID. Ground truth instance IDs are assigned to patches by searching the nearest matching instance point from annotated instance point clouds. For each 2D or 3D instance, the majority ID of their patches determines the ground truth instance ID. The Merging Discriminator is trained with a binary classification loss, where the label is positive ($\mathcal{G} = 1$) if the 2D and 3D instances share the same ground truth instance ID, or negative ($\mathcal{G} = 0$) otherwise:

$$\mathcal{L}_{segm} = \frac{1}{J} \sum_{j=1}^{J} \sum_{k=1}^{K} \text{CrossEntropy}(\text{MergingDiscriminator}(\mathcal{O}_j^{2D}, \mathcal{O}_k^{3D}, D_{j,k}), \mathcal{G}_{j,k}). \tag{4}$$

The function MergingDiscriminator($\cdot$) is an MLP network which takes as input the 2D instance features $\mathcal{O}_j^{2D}$, 3D instance features $\mathcal{O}_k^{3D}$ and their Euclidean distance $D_{j,k}$, and outputs a 2-dimensional logit vector. After extensive pre-training, the function MergingDiscriminator($\cdot$) efficiently integrates 2D instances into existing 3D instances to maintain mult-view and geometrically consistent 3D representations that can be updated.

**Feature Distillation and Language Alignment for 3D Instances.** To align 3D instances with language semantics, we leverage contrastive learning on large-scale 3D-language pairs from SceneVerse [50] and g3D-LF [15]. Given a 3D instance feature $\mathcal{O}_i$ and its corresponding annotated language description feature $\mathcal{T}_i$ extracted from CLIP text encoder, we treat $\mathcal{T}_i$ as the positive sample

and descriptions of other instances serve as negatives:

$$\mathcal{L}_{instance\_text} = \frac{1}{I}\sum_{i=1}^{I}\text{CrossEntropy}(\{\text{CosSim}(\mathcal{O}_i, \mathcal{T}_j)/\tau\}_{j=1}^{J}, i). \quad (5)$$

However, the generalization ability is limited by the scale of 3D-language data remains substantially smaller than that of image-language datasets: millions vs. billions [11]. We thus further enhance generalization by distilling visual knowledge from CLIP [11] into our Dynam3D model:

$$\mathcal{L}_{instance\_distillation} = \frac{1}{I}\sum_{i=1}^{I}\text{CrossEntropy}(\{\text{CosSim}(\mathcal{O}_i, \mathcal{O}_j^{gt})/\tau\}_{j=1}^{J}, i). \quad (6)$$

To obtain the ground-truth instance feature $\mathcal{O}_i^{gt}$ for distillation, we apply FastSAM to generate 2D instance masks and adopt the Masked Average Pooling (MAP) strategy from Feature Splatting [51] to average pool patch-level features within each instance mask and obtain $\mathcal{O}_j^{gt}$. However, we observe that the instance-level features extracted in this strategy are interfered by noise from the overall image background. The ground-truth instance features of the same 3D instance obtained from different views exhibit a significant gap, which greatly affects the effectiveness of distillation since one of our goals is to achieve multi-view consistency in the representation of 3D instances. Consequently, we propose a strategy of Subspace Contrastive Learning:

$$\mathcal{L}_{subspace\_distillation} = \frac{1}{I}\sum_{i=1}^{I}\text{CrossEntropy}(\{\text{CosSim}(\,(\mathcal{O}_i - \mathcal{V}_j), (\mathcal{O}_j^{gt} - \mathcal{V}_j)\,)/\tau\}_{j=1}^{J}, i), \quad (7)$$

where $\mathcal{V}_j$ is computed by average pooling all patch features within the given 2D view to yield the local semantic center of this view, *i.e.* semantic subspace. In Equation 6, instance features are optimized by maximizing cosine similarity with respect to the origin of the CLIP semantic space as the anchor. As a result, positive samples are pulled closer and negative samples are pushed farther apart. However, ground truth bias of different views can impede this contrastive process. In Equation 7, we replace the origin anchor with semantic center $\mathcal{V}_j$ of the view to mitigate the bias effect, impose a stronger optimization constraint and promote a sparser feature space with improved representational capacity.

**Feature Distillation and Language Alignment for 3D Zones.** As shown in Figure 2 and Figure 3, we introduce the zone-level representations $\mathcal{Z}$ to further capture coarse-grained spatial layout context. Specifically, our Dynam3D partitions the 3D world coordinate space into uniform cubic zones (each spanning several cubic meters) and employs a zone encoder to aggregate the instance-level features $\mathcal{O}$ within each zone to obtain $\mathcal{Z}$. The encoding process is similar to Equation 3. For feature distillation at the zone level, our Dynam3D adopts a relatively simple strategy: it uses a zone encoder to aggregate 3D instances that belong to the same 2D view, and then aligns the aggregated zone representation $\mathcal{Z}$ with the CLIP feature of the entire 2D view. Although the aggregated instances do not strictly come from the same cube zone, this approach ensures the quality of the distilled ground-truth features. For zone-level language alignment, we follow g3D-LF [15] to use Fine-grained Contrastive Learning for long-text contrastive supervision. Specifically, we compute an affinity matrix between the instance representations within a zone and the long-text representations to measure similarity, and then perform contrastive learning across different zones and texts.

## 3.2 3D Vision-Language Model for Navigation

As illustrated in Figure 2, Dynam3D constructs hierarchical 3D representations, spanning from fine-grained object instances to large-scale environmental zones. Leveraging these multi-level 3D representations as perceptual inputs, we introduce a dedicated 3D Vision-Language Model (3D-VLM) tailored for VLN tasks.

**Encoding Panoramic 3D Patch Tokens via Generalizable Feature Fields.** To effectively capture fine-grained geometric and semantic information within the surrounding panorama of the agent, we build upon the approach of g3D-LF [15] and adopt a generalizable feature field model to predict agent-centric 3D patch tokens. Specifically, we uniformly sample $12\times48$ rays covering a $90°$ vertical and $360°$ horizontal field-of-view around the agent, rendering both the 3D patch features $\hat{\mathbf{g}}$ and their corresponding depth estimates. These features with positional embeddings provide rich and spatially grounded representations of the scene geometry and semantics from the egocentric viewpoint.

**Multimodal Reasoning and Action Prediction.** To balance multimodal reasoning capabilities with computational efficiency, the 3.8 billion-parameter LLaVA-Phi-3-mini [52, 53] is integrated into

the proposed 3D-VLM framework. Since the 3D tokens (patch-instance-zone) are aligned with the semantic space of CLIP-ViT-L/14@336px [11], the strong multimodal understanding and reasoning abilities of this 2D-VLM can be effectively transferred to the 3D domain.

As shown in Figure 2, the input and output format of our 3D-VLM is:

**Input:** $< user > \{patch\_tokens\}\{instance\_tokens\}\{zone\_tokens\}\{instruction\_tokens\}$ $\{history\_action\_tokens\} < end >< assistant >$

**Output:** Next action: 1) Turn left $\theta$ degree. 2) Turn right $\theta$ degree. 3) Forward $d$ cm. 4) Stop.

*<user>* is a special token in LLaVA [36] used to indicate that the following tokens are context. *<end>* marks the end of a sequence.*<assistant>* indicates that the following tokens are the response of the model. To encode the relative positional relationship between 3D tokens and the agent, the relative coordinates $[x_c, y_c, z_c]$, *i.e.* camera coordinates of each 3D token to the agent are calculated along with the relative distance $D_c$ and the relative horizontal angle $\theta_c$. $[x_c, y_c, z_c, D_c, cos(\theta_c), sin(\theta_c)]$ of each token are then fed into a MLP network to generate the corresponding positional embeddings.

The 3D patch tokens $\{patch\_tokens\}$ rendered from the generalizable feature field are organized in a row-major order of $12\times48$ tokens, starting from the rays directly behind the agent and proceeding clockwise. This strategy is similar to that used in the pre-trained LLaVA-Phi-3-mini model [52, 53] when handling a single-view image. The instance tokens $\{instance\_tokens\}$ and zone tokens $\{zone\_tokens\}$ are sorted by their Euclidean distance to the agent from nearest to farthest. As shown in Figure 2, 3D-VLM outputs atomic actions with turning angles or movement distances. The history actions $\{history\_action\_tokens\}$ store the four most recent action texts, padding with the special token $< none >$ if fewer than four are available.

## 4 Experiments

### 4.1 Comparison with SOTA Methods

As shown in Tables 1 and 2, we evaluate the navigation performance of our Dynam3D across three distinct continuous-environment VLN benchmarks. Specifically, the R2R-CE dataset (Tables 1) provides step-by-step and following instructions. Compared to prior state-of-the-art methods, *e.g.*, g3D-LF and Uni-NaVid, our Dynam3D achieves an improvement of nearly 5% in navigation success rate (SR). Furthermore, despite the utilization of a large model, our Dynam3D maintains a smaller parameter footprint (3.8B vs. 7B) relative to the video-based Uni-NaVid. This highlights the superior efficiency of our model.

To ensure a fair comparison on the more challenging and realistic benchmarks such as REVERIE-CE which use coarse-grained and high-level destination description, and NavRAG-CE which requires understanding complex user demands, we retrain NaVid and g3D-LF on our training dataset and evaluate on these two benchmarks (Table 2). Our Dynam3D still demonstrates substantial improvements, outperforming NaVid by over 13% in Success Rate (SR) on REVERIE-CE and by over 5% on NavRAG-CE. The detailed experimental setup can be found in the supplementary materials.

Table 1: Evaluation of VLN on R2R-CE with monocular setting. ∗ denotes zero-shot method.

| Methods | LLM | Scene Representation | R2R-CE Val | | | | R2R-CE Test | | | |
|---|---|---|---|---|---|---|---|---|---|---|
| | | | NE↓ | OSR↑ | SR↑ | SPL↑ | NE↓ | OSR↑ | SR↑ | SPL↑ |
| CM$^2$ [54] | × | Semantic Map | 7.02 | 41.5 | 34.3 | 27.6 | 7.7 | 39 | 31 | 24 |
| WS-MGMap [55] | × | Multi-Granularity Semantic Map | 6.28 | 47.6 | 38.9 | 34.3 | 7.11 | 45 | 35 | 28 |
| InstructNav∗ [56] | ✓ | Semantic Value Map | 6.89 | - | 31 | 24 | - | - | - | - |
| AO-Planner∗ [57] | ✓ | Visual Affordance Prompts | 6.95 | 38.3 | 25.5 | 16.6 | - | - | - | - |
| NaVid [5] | ✓ | Video Frames | 5.47 | 49.1 | 37.4 | 35.9 | - | - | - | - |
| VLN-3DFF [28] | × | Feature Fields | 5.95 | 55.8 | 44.9 | 30.4 | 6.24 | 54.4 | 43.7 | 28.9 |
| g3D-LF [15] | × | Feature Fields | 5.70 | 59.5 | 47.2 | 34.6 | 6.00 | 57.5 | 46.3 | 32.2 |
| Uni-NaVid [6] | ✓ | Multi-Granularity Video Frames | 5.58 | 53.3 | 47.0 | 42.7 | - | - | - | - |
| Dynam3D (Ours) | ✓ | 3D Patch-Instance-Zone Tokens | 5.34 | 62.1 | 52.9 | 45.7 | 5.53 | 60.4 | 51.4 | 44.8 |

### 4.2 Experiments on Pre-exploration and Lifelong Memory

As shown in Table 3, we additionally evaluate the performance under the *Pre-exploration* and *Lifelong Memory* settings to further demonstrate the advantages of our Dynam3D. The pre-explored panoramic

Table 2: Evaluation of VLN on REVERIE-CE and NavRAG-CE with monocular setting. ∗ denotes zero-shot method.

| Methods | LLM | Scene Representation | REVERIE-CE Val | | | | NavRAG-CE Val | | | |
|---|---|---|---|---|---|---|---|---|---|---|
| | | | NE↓ | OSR↑ | SR↑ | SPL↑ | NE↓ | OSR↑ | SR↑ | SPL↑ |
| InstructNav∗ [56] | ✓ | Semantic Value Map | 7.44 | 31.5 | 25.2 | 19.1 | 9.83 | 24.1 | 17.4 | 10.9 |
| NaVid [5] | ✓ | Video Frames | 6.74 | 36.3 | 26.6 | 20.8 | 9.35 | 29.6 | 19.4 | 13.9 |
| g3D-LF [15] | × | Feature Fields | 6.50 | 41.6 | 34.4 | 23.8 | 8.85 | 31.8 | 21.4 | 13.5 |
| Dynam3D (Ours) | ✓ | 3D Patch-Instance-Zone Tokens | **6.22** | **48.9** | **40.1** | **28.5** | **8.12** | **38.4** | **24.7** | **18.8** |

images from the Pre-exploration setting are collected at the navigable viewpoints annotated in the Matterport3D [22] dataset, which are then used to construct the Patch-Instance-Zone representations of the entire scene. For the Lifelong Memory setting, we group the evaluation episodes by scene with navigation samples from the same scene evaluated consecutively within a group. For each scene, previously stored 3D representations can be leveraged in subsequent episodes to simulate gradual familiarization of the agent with the environment during task execution.

Table 3 shows that the Pre-exploration strategy enables our Dynam3D to achieve over a 5% improvement in Success Rate (SR) on R2R-CE and an 8% improvement on REVERIE-CE. Under the Lifelong Memory setting, our Dynam3D also achieves performance gains, with a 2.7% SR improvement on R2R-CE and a 4.9% SR improvement on REVERIE-CE. Compared to NaVid [5] which uses a video-based large model, our Dynam3D employing both the Pre-exploration and Lifelong Memory achieves over a 20% increase in navigation success rate (SR).

Table 3: Evaluation of VLN for Pre-exploration and Lifelong Memory. **Pre-exploration** allows agents to scan and encode environmental representations before evaluation, while **Lifelong Memory** enables agents to retain the environmental representations of previous episodes for subsequent episodes.

| Methods | Pre-exploration | Lifelong Memory | R2R-CE Val | | | | REVERIE-CE Val | | | |
|---|---|---|---|---|---|---|---|---|---|---|
| | | | NE↓ | OSR↑ | SR↑ | SPL↑ | NE↓ | OSR↑ | SR↑ | SPL↑ |
| NaVid [5] | × | × | 5.47 | 49.1 | 37.4 | 35.9 | 6.74 | 36.3 | 26.6 | 20.8 |
| g3D-LF [15] | × | × | 5.70 | 59.5 | 47.2 | 34.6 | 6.50 | 41.6 | 34.4 | 23.8 |
| g3D-LF [15] | ✓ | ✓ | 5.46 | 62.5 | 51.8 | 39.9 | 6.44 | 43.3 | 37.1 | 25.9 |
| Dynam3D (Ours) | × | × | 5.34 | 62.1 | 52.9 | 45.7 | 6.22 | 48.9 | 40.1 | 28.5 |
| Dynam3D (Ours) | ✓ | × | **5.04** | 66.2 | 57.1 | **52.7** | 6.09 | **56.8** | 48.1 | 37.3 |
| Dynam3D (Ours) | × | ✓ | 5.21 | 64.4 | 55.6 | 48.1 | 6.31 | 52.8 | 45.0 | 32.7 |
| Dynam3D (Ours) | ✓ | ✓ | 5.11 | **67.2** | **58.4** | 50.4 | **6.02** | 56.4 | **49.5** | **38.1** |

## 4.3 Experiments on Real World and Dynamic Environment

As shown in Tables 4, 5 and Figure 4, we evaluate our Dynam3D on both real-world static and dynamic environments using the Hello Robot Stretch 3. Each setting includes 20 test cases, and navigation is deemed successful if the robot stops within 1 meter of the target. In the static environment (Table 4) Dynam3D achieves a 20% higher success rate than baselines, reaching 70% after pre-exploration. In the dynamic setting (Figure 4 and Table 5), the target is manually moved to another location once the robot reach within two meters of the original target. our Dynam3D consistently outperforms all baselines, demonstrating strong robustness to environmental changes. The detailed experimental setup can be found in the supplementary materials.

Table 4: Real-world navigation experiments in **static** environments.

| Methods | NE↓ | OSR↑ | SR↑ |
|---|---|---|---|
| NaVid | 2.2 | 45 | 35 |
| g3D-LF | 3.1 | 40 | 30 |
| Dynam3D | 1.4 | 65 | 55 |
| + Pre-exploration | 0.8 | 75 | 70 |

Table 5: Real-world navigation experiments in **dynamic** environments.

| Methods | NE↓ | OSR↑ | SR↑ |
|---|---|---|---|
| NaVid | 3.6 | 45 | 20 |
| g3D-LF | 4.6 | 35 | 10 |
| Dynam3D | 1.9 | 60 | 45 |
| + Pre-exploration | 1.4 | 75 | 45 |

## 4.4 Analysis for Dynamic Representation and Lifelong Memory

In our Dynam3D, the agent discards outdated representations within its current field-of-view and adds or updates new ones to ensure real-time understanding of dynamic environments with each new observation. However, this does not conflict with its advantage of maintaining lifelong memory.

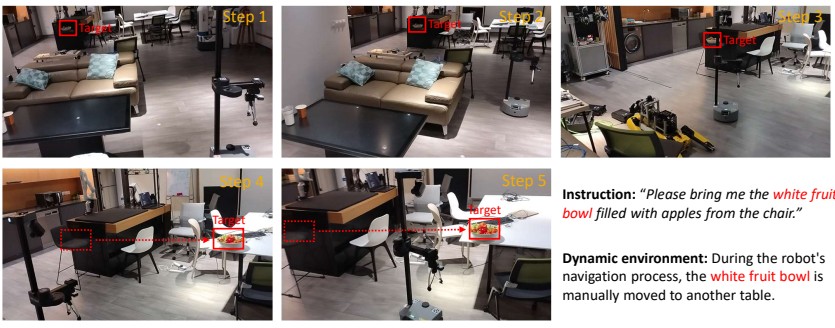

Figure 4: A demonstration of navigation in a dynamic real-world environment.

Features corresponding to regions outside the robot's current view are still preserved and not updated, and serve as long-term memory to support navigation. This is due to the limited field of view of the monocular camera, where the majority of the scene is preserved since it lies outside the current view of the robot. These features are retained until new observations covering those regions are available to update them. This highlights the ability of our Dynam3D to balance real-time adaptability in dynamic environments with the effective use of long-term memory.

## 4.5 Computational Cost and Real-Time Analysis

We evaluate computational cost on the R2R-CE dataset using a single NVIDIA RTX 4090 GPU. During training, each navigation step takes 455ms (∼0.46 seconds) on average: 83ms for 3D representation updates, 315ms for large language model, and 57ms for other operations. During inference, the average step time increases to 649ms (∼0.65 seconds) with 83ms for 3D representation updates, 540ms for large language model inference, and 26ms for the remaining components. Most navigation episodes can be completed within 20 to 40 navigation steps, our navigation system supports real-time 3D representation updates and navigation action prediction for efficient training and inference.

## 4.6 Ablation Study

Table 6: Ablation Study of Dynam3D on R2R-CE and REVERIE-CE Val Unseen benchmarks. We analyze the contribution of different components: input view type (currently observed **Single View** patches vs. **Rendered Panoramic** patches), High-level 3D representations (**Instance** and **Zone**), and the **Subspace Align.** loss used for supervising instance representations.

| Single View | Rendered Pano | Instance | Subspace Align. | Zone | R2R-CE Val Unseen | | | | REVERIE-CE Val Unseen | | | |
|---|---|---|---|---|---|---|---|---|---|---|---|---|
| | | | | | NE↓ | OSR↑ | SR↑ | SPL↑ | NE↓ | OSR↑ | SR↑ | SPL↑ |
| ✗ | ✗ | ✓ | ✓ | ✓ | 7.59 | 43.9 | 32.7 | 21.1 | 8.76 | 27.4 | 19.3 | 11.4 |
| ✓ | ✗ | ✓ | ✓ | ✓ | 6.14 | 51.7 | 43.1 | 33.9 | 7.85 | 35.9 | 28.4 | 21.8 |
| ✗ | ✓ | ✗ | ✗ | ✗ | 5.63 | 51.1 | 45.7 | 40.2 | 7.89 | 34.8 | 25.7 | 17.8 |
| ✗ | ✓ | ✗ | ✗ | ✓ | 5.77 | 54.7 | 47.9 | 43.3 | 6.58 | 38.5 | 26.8 | 21.0 |
| ✗ | ✓ | ✓ | ✓ | ✗ | **5.26** | 61.8 | 52.4 | 45.7 | 6.37 | 46.2 | 39.3 | 26.2 |
| ✗ | ✓ | ✓ | ✗ | ✓ | 5.44 | 58.8 | 50.7 | 43.2 | 6.31 | 45.1 | 38.4 | 25.8 |
| ✗ | ✓ | ✓ | ✓ | ✓ | 5.34 | **62.1** | **52.9** | **45.7** | **6.22** | **48.9** | **40.1** | **28.5** |

**Contribution of 3D Representations.** From the full model (last row), removing the rendered panoramic patch tokens (row 1), instance tokens (row 4), zone tokens (row 5), or both instance and zone tokens (row 3) leads to performance drops of varying degrees. The results show that the impact on performance follows the order: Patch > Instance > Zone tokens.

Specifically, when both 3D instance and zone representations are removed and only 3D patch tokens from the feature fields [15] are used (row 3), there is a substantial performance drop, particularly on REVERIE-CE, where SR decreases by nearly 15%. This highlights the critical role of the hierarchical instance-zone representation in supporting effective navigation and large-scale exploration, as local patch-level features alone provide limited spatial coverage.

**Importance of Subspace Alignment.** Without Subspace Alignment supervision (row 6), the navigation performance significantly decreases. This highlights the limitations of naive CLIP feature distillation for 3D instance supervision. Subspace Contrastive Learning effectively mitigates the instance feature bias from different views.

**Value of Rendered Panoramic Views.** As analyzed in VLN-3DFF [28], rendered panoramic patch tokens substantially mitigate the limited field of view caused by monocular cameras on most robots. Comparing row 2 and last row in Table 6, replacing the rendered panoramic tokens with CLIP patch features extracted from the current monocular view results in about a 10% drop in navigation success rates on both R2R-CE and REVERIE-CE.

Table 7: Robustness study on the R2R-CE Val Unseen benchmark with the simulated SLAM noise and depth noise.

| SLAM Noise | Depth Noise | NE↓ | OSR↑ | SR↑ | SPL↑ |
|:---:|:---:|:---:|:---:|:---:|:---:|
| ✗ | ✗ | 5.34 | 62.1 | 52.9 | 45.7 |
| ✓ | ✗ | 5.46 | 61.3 | 52.1 | 45.0 |
| ✗ | ✓ | 5.39 | 61.9 | 52.3 | 45.2 |
| ✓ | ✓ | 5.47 | 60.7 | 50.9 | 44.1 |

**Robustness to Noise.** To further validate the model's navigation performance in noisy environments, we introduce different types of noise into the simulator and evaluate Dynam3D's performance. We simulate the potential noise from a SLAM system in localization and pose estimation, as well as the measurement errors inherent in depth cameras.

To simulate SLAM noise, we introduce localization noise, uniformly sampled from -5 cm to +5 cm, and orientation noise, uniformly sampled from -3 degrees to +3 degrees, to the agent's position at each simulation step. To simulate depth camera noise, we add random noise to the depth maps acquired from the simulator. For depths within 2 meters, noise is sampled between -4 cm and +4 cm; for depths from 2 to 3 meters, noise is sampled between -6 cm and +6 cm; for depths over 3 meters, noise is sampled between -10 cm and +10 cm.

As shown in Table 7, even with the simultaneous addition of simulated SLAM and depth noise, the navigation success rate (SR) only decreased by approximately 2% (comparing the last row to the first row), demonstrating Dynam3D's robustness to noise. A significant reason is that our Dynam3D leverages 2D foundation models (e.g., CLIP [11], FastSAM [12]) to extract visual semantic features, which are unaffected by SLAM and depth noise. Even with minor spatial position deviations due to noise, the 3D-aware representations obtained by our model through the hierarchical encoding of Patch-Instance-Zone remain sufficient for perceiving spatial geometry and scene layout at the object and zone levels.

## 5 Conclusion

We introduce Dynam3D, a dynamic hierarchical 3D representation framework for monocular vision-and-language navigation. By aligning patch-instance-zone features with language semantics and enabling real-time scene updates, our Dynam3D enhances spatial understanding, long-term memory, and adaptability in dynamic environments. Our model achieves state-of-the-art results on multiple VLN benchmarks and demonstrates strong generalization in real-world deployment. These results highlight the value of structured and dynamically updated 3D representations for embodied navigation.

**Limitations.** Our Dynam3D predicts navigation actions without explicitly outputting the coordinate of target instance, limiting its applicability to some tasks such as mobile manipulation. Moreover, it lacks capabilities for question answering, dialogue, and task updates, showing potential directions for better navigation agents.

## Acknowledgement

This research work is supported by the Agency for Science, Technology and Research (A*STAR) under its MTC Programmatic Funds (Grant No. M23L7b0021), and the Tier 2 grant MOE-T2EP20124-0015 from the Singapore Ministry of Education.

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

# A  Supplementary Material

## A.1   Datasets and Experimental Details

**3D-Language Datasets and Training Details.** To train the Dynam3D representation model, we follow SceneVerse [50] and g3D-LF [15] in collecting over 5K scenes with 2M language annotations from ScanNet [47], HM3D [48], Matterport3D [22], 3RScan [49], ARKitScenes [58], and Structured3D [59]. For each episode, we randomly sample sufficient posed RGB-D images from raw videos or the Habitat simulator [23] to construct and update the hierarchical patch-instance-zone representations. The updated representations after each observed frame is supervised using the losses defined in Section 3.1. We pre-train our Dynam3D representation model on the aforementioned dataset for 100K episodes (approximately 8 days) using four RTX 6000 Ada GPUs. The training is performed with a batch size of 4 and a learning rate of 1e-4.

**Navigation Datasets and Training Details.** To train our 3D-VLM with sufficient navigation data, we transfer datasets generated by ScaleVLN [60] and NavRAG [4] from discrete environments to the continuous Habitat simulator [23]. After removing samples with impassable paths, we obtain 4M+ instruction-trajectory pairs in continuous settings. For a comprehensive and fair evaluation, we evaluate our model on R2R-CE [3], REVERIE-CE and NavRAG-CE by transferring REVERIE [2] and NavRAG [4] datasets to continuous environments. To balance data quality and scale, we randomly sample model-generated data (ScaleVLN, NavRAG; 4M+) and human-annotated data (R2R-CE, REVERIE-CE; 20K+) at a 1:1 ratio during 3D-VLM training. Navigation training proceeds in two stages: 1) **Imitation learning.** The agent strictly follows ground-truth paths to enhance instruction following and multimodal alignment; 2) **Exploration and correction.** Following ETPNav [26], we adopt a waypoint predictor [24] to generate multiple candidate waypoints. We utilize the DAgger strategy [61, 18] to enhance error correction by deliberately introducing probabilistic deviations that mislead the agent towards incorrect waypoints. The agent is then guided back to the correct path, thereby strengthening its ability to recover from navigation errors. We pre-train the 3D-VLM model on the navigation datasets for 100K episodes (50K for stage one, 50K for stage two, approximately 9 days) using two RTX 6000 Ada GPUs. The training is performed with a batch size of 4 and a learning rate of 1e-6. During training, all parameters of the 3.8B LLaVA-Phi-3-mini [52, 53] are optimized, except the generalizable feature field model [15] and the pre-trained Dynam3D representation model. To mitigate memory consumption and enable efficient training of large models, we employ the Adafactor optimizer [62] in conjunction with Gradient Checkpointing [63].

**Details of Real-world Navigation.** We employ the Hello Robot Stretch 3 for real-world navigation experiments, leveraging its real-time localization and pose estimation capabilities. An Intel RealSense D435i RGB-D camera is mounted on the robot's head to facilitate 3D scene representation construction and incremental updates. Our real-world experimental framework is adapted from DynaMem [64], with extensions for obstacle avoidance and movement. The model is deployed on a workstation equipped with an NVIDIA RTX 4090 GPU and 64GB of RAM, and communicates with the robot over a local area network established via a WiFi access point. The experimental environment consists of a home-style setting constructed for robot evaluation, encompassing a living room, kitchen, meeting room, and office. To ensure a fair comparison under the unseen setting, none of the objects or rooms within the environment are included in the training data.

