# OpenReview forum: "Dynam3D: Dynamic Layered 3D Tokens Empower VLM for Vision-and-Language Navigation"
_NeurIPS.cc/2025/Conference — NeurIPS 2025 oral_

### Official Review · Reviewer_CAAU · 2025-06-17

**Clarity:** 2
**Significance:** 3
**Originality:** 2
**Rating:** 5
**Confidence:** 3

**Summary:**

This paper introduces Dynam3D, a system designed to help robots follow natural language instructions and navigate through indoor environments using only a single RGB-D camera. Unlike previous methods that rely on video-based models, Dynam3D builds a dynamic 3D representation of the environment using a layered structure that includes small visual patches, object instances, and larger spatial zones. It updates this representation in real time, allowing the system to better understand the 3D layout, remember past observations, and adapt to changes like moving objects. The authors also introduce a 3D Vision-Language Large Model  to predict the action in a continuous environment. Experiments show that Dynam3D achieves strong results on several navigation benchmarks and works well in real-world and dynamic settings.

**Questions:**

How would the performance be affected if the camera parameters are miscalculated or the depth data contains noise?

Compared with discrete VLN settings, what are the strengths and limitations of VLN in continuous environments? It is advisable to clarify this in the paper.

**Ethical Concerns:**

["NO or VERY MINOR ethics concerns only"]

**Final Justification:**

The authors’ rebuttal addresses most of my concerns, so I am increasing my score. Although the pipeline remains rather hand-crafted and complex—which may limit its broader applicability and weigh on the evaluation—I still consider the work suitable for acceptance as a conference paper.

**Limitations:**

The authors have not adequately discussed the application scenarios of the algorithm and the requirements for data quality.

**Quality:**

3

**Strengths And Weaknesses:**

**Strengths**

The hierarchical 3D representation enables navigation in dynamic scenes by updating only the instance-level features, allowing the algorithm to adapt its behavior in response to moving objects.

The authors applied this method to a real robot and demonstrated its performance in real-world operation.

**Weaknesses**

The reviewer thinks the biggest weakness of this paper is the lack of a comprehensive ablation study. For example, while the method outperforms previous approaches in the experiments, it's unclear whether the comparison was made under the same model parameter size. In real-world applications, RGB-D cameras produce different types of noise depending on the scene, and the impact of such depth noise on navigation has not been sufficiently analyzed. Moreover, the point clouds shown in the paper appear almost perfect, so the reviewer has reservations about how well the algorithm would perform in real-world scenarios with noisy data.

The pipeline of this algorithm is highly hand-crafted, which may lead to poor robustness in handling certain edge cases, such as camera tracking errors or segmentation mistakes that occur during intermediate stages of the pipeline.

---

> ### Author Rebuttal · Authors · 2025-07-30
>
> Thanks for your valuable comments.
> ## **1. Model Parameter Size Analysis**
> As shown in Section 4.1, our Dynam3D has an advantage in terms of parameter size compared to major large model-based VLN methods: **NaVid 7B, Uni-NaVid 7B, and Dynam3D 3.8B**. As shown in Tables 1 and 2 of the original submission, our Dynam3D achieves stronger performance with the smaller parameter scale.
> For more ablation study analysis to more clearly demonstrate the performance contribution of each component in Dynam3D, please refer to our response to Reviewer QXMr.
>
> ## **2. Robustness Analysis to Noise**
> Thanks for your insightful suggestions. As demonstrated by the real-world experiments in Section 4.3 of the original submission, our Dynam3D exhibits excellent performance in real-world navigation based on the Stretch 3 robot. This reflects the robustness of our Dynam3D in real-world environments. A **ROS Nav2-based SLAM system** is developed for **precise localization and pose estimation** on the Stretch 3 robot by combining sensory data from **LiDAR, IMU, and a depth camera**. Although no official data is available, user reports and our rough estimates suggest a **positioning accuracy of approximately ±5 cm and an orientation accuracy of about ±3 degrees**.
> For the **Intel RealSense D435i depth camera** used in Stretch 3 robot, its official specifications indicate a depth error of **approximately ±4 cm within a 2-meter range and ±6 cm within a 3-meter range**.
> To further validate the practical impact of this noise on navigation performance, we add different types of noise to the simulator and evaluate the performance of our Dynam3D on R2R-CE Val Unseen in noisy environments.  In order to simulate SLAM noise, we introduce localization noise, uniformly sampled from -5 cm to +5 cm, and orientation noise, uniformly sampled from -3 degrees to +3 degrees, to the agent's position at each simulation step. To simulate depth camera noise, we add random noise to the depth maps acquired from the simulator. For depths within 2 meters, noise is sampled between -4 cm and +4 cm; for depths at 2~3 meters, noise is sampled between -6 cm and +6 cm; for depths over 3 meters, noise is sampled between -10cm and +10 cm.
> | SLAM Noise | Depth Noise | NE$\downarrow$ | OSR$\uparrow$ | SR$\uparrow$ | SPL$\uparrow$ |
> |---|---|---|---|---|---|
> | × | × | 5.34 | 62.1 | 52.9 | 45.7 |
> | √ | × | 5.46 | 61.3 | 52.1 | 45.0 |
> | × | √ | 5.39 | 61.9 | 52.3 | 45.2 |
> | √ | √ | 5.47 | 60.7 | 50.9 | 44.1 |
>
> As shown in the table above,**even with the simultaneous addition of simulated SLAM noise and depth noise, the navigation success rate (SR) only decreased by approximately 2%** (comparing the last row to the first row), demonstrating Dynam3D's robustness to noise. A significant reason is, unlike point cloud-based 3D-VLM models require near-perfect point clouds for their point cloud encoder to extract semantics based on texture and fine geometry, **our Dynam3D leverages 2D foundation models (e.g., CLIP, FastSAM) to extract visual semantic features (patch-level), which are unaffected by SLAM and depth noise**. Our Dynam3D acquires 3D-aware representations through a hierarchical encoding of patch-instance-zones, even with minor spatial position deviations due to noise, this strategy remains sufficient for perceiving spatial geometry and scene layout at the object and zone levels.
>
> ## **3. Comparing VLN in Discrete and Continuous Environments**
> In the discrete environment VLN, a predefined navigable connectivity graph with nodes is used. The agent can only move between these nodes in the graph and is typically equipped with a panoramic RGB-D camera. Due to the limitations of the computational resources and model capacity in few years ago, discrete environments significantly simplify the navigation process for easier research. For example, it eliminates the need for low-level action prediction, ignores collision avoidance, and disregards potential environmental changes.
>
> However, a significant drawback is that navigation models trained in discrete environments are almost impossible to deploy on real robots due to two main reasons: 1) Real-world environments rarely have perfectly predefined navigation connectivity graph, and 2) The vast majority of robots do not possess panoramic RGB-D cameras due to their large size and high cost.
> In contrast, continuous environments (VLN-CE) , especially with a monocular camera setting, show greater practical value. Although VLN model performance on continuous environment benchmarks (R2R-CE, REVERIE-CE) has not yet surpassed that of discrete environments (R2R, REVERIE), the gap for real-world deployment is significantly reduced. The continuous settings require models to predict low-level actions, e.g. rotation and forward movement, and handle collision avoidance. **What's more, continuous VLN largely mitigates a key limitation, i.e. discrete VLN models tend to overfit to perfect connectivity graphs, which severely restricts their generalization in real-world navigation**.
>
> ## **4. Discussion for the application scenarios of Dynam3D and the requirements for data quality.**
> Dynam3D is designed for language-guided indoor navigation tasks carried out by robots equipped with precise SLAM and monocular RGB-D cameras, and is capable of adapting to real-time human and object movements.
>
> For data quality, posed RGB-D images are required as model input, the specific accuracy requirements for depth images and camera poses can be found in the **Robustness Analysis to Noise** section.

---

> > ### Comment · Reviewer_CAAU · 2025-08-05
> >
> > Thank you for the clarifications and additional experiments—they resolve most of my concerns.

---

### Official Review · Reviewer_kSWU · 2025-06-27

**Clarity:** 3
**Significance:** 3
**Originality:** 3
**Rating:** 5
**Confidence:** 4

**Summary:**

This paper proposes a new 3D hierarchical representation structure for organizing information in vision-language action (VLA) models. The core idea is inspired to some extent by mapping-based methods, but it also departs from them in important ways. The proposed structure acts as a sliding-window-style, multi-level, dynamically updated, fused, and removed semantic memory, rather than a persistent global map. This design addresses key issues faced by video-based VLA models, namely, shallow physical understanding and inefficient memory organization, as the authors themselves note in the intro second paragraph. Technically, the paper introduces a three-tier 3D representation and its corresponding update/fusion mechanism, and demonstrates its effectiveness through both simulator-based and real-world robot experiments.

**Questions:**

I would appreciate it if the authors could primarily address the two major weaknesses:

Generalization: Please consider adding cross-dataset evaluation or using previously unseen real-world environments (e.g., outdoor scenes) to verify the robustness of 3D-language alignment.

Inference Speed: It would be highly valuable if the authors could propose insightful and practical ways to reduce inference latency. If the inference can be sped up noticeably, it would significantly strengthen the paper.

If possible, it would also be great to see the authors touch on the remaining concerns—about demo quality, Section 4.2 clarity, and action format—even if only in discussion or qualitative analysis.

**Ethical Concerns:**

["NO or VERY MINOR ethics concerns only"]

**Final Justification:**

This is a fairly good piece of work, with no obvious flaws or weaknesses. I believe it meets the bar for NeurIPS acceptance, so I am keeping my original score.

**Limitations:**

Yes, the limitations are discussed.

**Paper Formatting Concerns:**

None.

**Quality:**

3

**Strengths And Weaknesses:**

Strengths

1、This is a strong paper with a clear motivation and solid technical contributions. The proposed direction feels like a natural and promising step forward given the current state of research, bringing 3D awareness into VLA without incurring excessive computational or memory overhead.

2、The technical design of this work is carefully crafted. It introduces a multi-level 3D representation structure, and incorporates dynamic mechanisms for updating, merging, and pruning tokens, which ensures computational efficiency and a certain degree of real-time capability. The technical components are well-formulated with clear mathematical definitions, allowing readers to precisely understand the authors' intentions.

3、The experiments are thorough, and the comparisons are both relevant and novel. The accompanying demo, though simple, is convincing. Importantly, the insights presented in this work could inspire further research in the community, it is a meaningful exploration.

Weaknesses

1、My main concern lies in generalization. Higher-dimensional representations typically require more data to train effectively. Existing VLA systems like Uni-NaVid are trained on massive 2D-scale datasets (millions to tens of millions of samples), whereas this work does not appear to increase the data volume in proportion to the added complexity of 3D representations. This raises the question of whether the proposed 3D-language alignment generalizes well, especially under real-world conditions with noise and the inherent sim-to-real gap. I strongly encourage the authors to conduct more experiments specifically targeting generalization, either within simulators (e.g., cross-scene or cross-dataset setups) or on real robots using unseen environments, such as outdoor scenes.

2、The second major issue is inference speed. As shown in Section 4.4, the real-world inference runs at under 2 FPS, which is somewhat slow for practical deployment. Interestingly, as noted in Section 4.1, the model's parameter size is actually smaller than other VLA baselines. This suggests that there may be unexplored opportunities—either at the model level or in engineering—to significantly improve runtime performance. I hope the authors can propose concrete and actionable ideas to reduce inference latency.

3、A third point is that the robot demo leaves room for improvement. Given the introduction of 3D spatial understanding, one would expect the system to showcase unique VLN capabilities that go beyond existing 2D-based models. More spatially grounded or fine-grained instructions could have made the demo more impressive. As it stands, it feels somewhat ordinary.

(Minor concern) The intent of Section 4.2 is a bit unclear. The proposed representation is described as "ephemeral and disposable," yet the experiments on pre-exploration and lifelong memory imply persistent memory reuse. It would be helpful for the authors to clarify this apparent contradiction and explicitly state the purpose of these experiments.

(Minor concern) Regarding the action prediction format in VLA models—have the authors considered using more advanced policy structures, such as multi-step forecasting or diffusion-based policies? Given the richness of the spatial representation, such approaches could lead to more foresightful and smoother navigation.

---

> ### Author Rebuttal · Authors · 2025-07-30
>
> Thanks for your insightful comments and suggestions!
> ## **1. Analysis of Dynam3D's Generalization Capability**
> ### **1) 3D Representation Generalization**
> As detailed in the Supplementary Material (Line 503 of the original submission), we follow SceneVerse and g3D-LF to collect over 5K scenes with 2M language annotations from ScanNet, HM3D, Matterport3D, 3RScan, ARKitScenes, and Structured3D. This data is used to train our Dynam3D representation model and retrain the g3D-LF feature field model. HM3D, Matterport3D, and Structured3D are from simulators or renderers. The remaining training data ScanNet, 3RScan, and ARKitScenes provide over 2.5K scenes captured using real RGB-D cameras in real-world environments. This helps eliminate the sim-to-real gap and ensures strong generalization of the learned representations to real-world scenarios.
>
> ### **2) VLN Model Generalization**
> As detailed in the supplementary materials (Line 503 of the original submission), we conducted mixed training across 861 environments (HM3D, MP3D) to enhance the navigational generalization of our Dynam3D. This involved over 4 million navigation samples, comprising both model-generated data (ScaleVLN, NavRAG; 4M+) and human-annotated data (R2R-CE, REVERIE-CE; 20K+). This diverse dataset included various navigation instructions, such as step-by-step following, high-level description, and user-demand instructions.
> To further validate the generalization performance of our Dynam3D, we conduct zero-shot evaluations on the **HM3D-OVON** Val Unseen dataset for open-vocabulary object navigation to show the cross-dataset performance and zero-shot generation:
> | Methods   | SR   | SPL                              |
> | --------- | ---- |:--------------------------------:|
> | VLFM      | 35.2 | 19.6                             |
> | Uni-NaVid | 39.5 | 19.8                             |
> | Dynam3D   | **42.7** | **22.4** |
>
> ## **2. Optimize inference latency with better action prediction policies**
> The dynamic 3D-VLM contexts arising from real-time updates of rendered panoramic patches, instance and zone tokens preclude the application of static-context inference acceleration techniques such as KV caching that is commonly employed in traditional video-based VLMs. However, as suggested by Reviewer kSWU, our Dynam3D has the potential to accelerate the navigation process using multi-step forecasting or diffusion-based policies by using high-quality 3D-aware representations. For example, a large 3D-VLM can execute **high-level encoding and reasoning at 2 FPS**. A **smaller action expert model** can use latent features from the 3D-VLM (similar to the VLA model π₀) as input to predict multiple consecutive navigation trajectory points at **over 20 FPS**. During robot movement, the **3D representation is continuously updated** in real time at **over 10 FPS**. Currently, our Dynam3D still uses text output for action prediction primarily to ensure a fair comparison with previous video-based models that also used text output. However, this navigation action output mode should only be a temporary solution. Once sufficient training data is available, smoother and faster navigation can be achieved with the better choice of predicting multi-step continuous trajectory points.
>
> ## **3. Better demos to showcase spatial reasoning ability**
> Thank you for the suggestion. As NeurIPS does not support media uploads this year, we are unable to include additional demos. Nevertheless, our Dynam3D exhibits clear advantages over video-based models in 3D perception, localization, and reasoning with better 3D-aware representations. If paper is accepted, we will provide more demonstrations in the camera-ready version to highlight its spatial reasoning capabilities and potential in object-centric tasks such as mobile manipulation, which require precise object localization and geometric understanding.
>
> ## **4. Dynamic Representation and Lifelong Memory in Section 4.2**
> Thank you for your suggestion. In our Dynam3D, the robot discards outdated representations within its current field-of-view and adds or updates new ones to ensure real-time understanding of dynamic environments with each new observation. However, this does not conflict with its advantage of maintaining lifelong memory. Features corresponding to regions outside the robot’s current view of the robot are still preserved and not updated, and serve as long-term memory to support navigation. This is due to the limited field of the monocular camera, where the majority of the scene is preserved since it lies outside the current view of the robot. These features are retained until new observations covering those regions are available to update them. This highlights the ability of our Dynam3D to balance real-time adaptability in dynamic environments with the effective use of long-term memory.

---

> > ### Comment · Reviewer_kSWU · 2025-08-02
> >
> > Thank you for your response, which has addressed my question.

---

### Official Review · Reviewer_QXMr · 2025-07-03

**Clarity:** 3
**Significance:** 3
**Originality:** 3
**Rating:** 5
**Confidence:** 5

**Summary:**

This paper proposes Dynam3D, a novel framework for vision-language navigation (VLN) that introduces a dynamic hierarchical 3D scene representation. The representation integrates patch, instance, and zone tokens, addressing challenges in spatial understanding, long-term memory, and flexibility to dynamic environments. The authors combine this representation with a pretrained VLM LLaVA-Phi-3-mini, and demonstrate superior performance across VLN benchmarks (R2R-CE, REVERIE-CE, NavRAG-CE). Furthermore, Dynam3D exhibits significant results in real-world navigation tasks, including static and dynamic environments.

**Questions:**

- I think the delivered message in the experiments of pre-exploration and lifelong memory (Section 4.2) can be polished. There is no doubt that pre-exploration would improve the performance, and it cannot highlight the advantage of Dynam3D since the comparison with other methods without pre-exploration is unfair. Instead, I think the proper message here is that Dynam3D makes lifelong memory feasible and thereby achieves a similar effect to pre-exploration.
- I would expect more details on the processing of VLN data, such as action merging, label smoothing or rebalancing. These details can be important to the final performance of VLN as demonstrated by prior works.
- An interesting and important question is how well the merging discriminator can perform. Are the merging results accurate? Providing some simple evaluations about the merging accuracy would offer more insights.

**Ethical Concerns:**

["NO or VERY MINOR ethics concerns only"]

**Final Justification:**

Technically solid and insightful paper. I recommend accept.

**Limitations:**

Yes

**Quality:**

3

**Strengths And Weaknesses:**

#### Strengths
- The methodology is well-grounded and technically sound. The idea of dynamically updated patch-instance-zone 3D representations is both innovative and well-motivated. And Dynam3D incorporates thoughtful designs for dynamic environments, including Frustum Culling and merging discriminator.
- The experimental results are significant, highlighting the strengths of Dynam3D in performing VLN tasks. In particular, Dynam3D shows strong performance in real-robot experiments.

#### Weaknesses
- I think the representation design is kind of overcomplicated. The pipeline for deriving various kinds of tokens involves complex procedures such as segmentation with FastSAM and rendering with g3D-LF. My concern persists despite the reported computational costs and real-time analyses. While it cannot be a major weakness to propose many dedicated and specific designs, it can make the method less appealing.
- Given such dedicated designs, I would expect more ablation results to figure out the contributions of different parts to the performance. For example, the contributions of 3D instance tokens and panoramic 3D patch tokens are not clearly reported. On the other hand, the patch, instance, and zone tokens represent different granularities. So my concern is that would such a heterogeneity confuse the model learning (e.g., can model distinguish their token types)?

---

> ### Author Rebuttal · Authors · 2025-07-30
>
> Thank you very much for your insightful comments! In response, we include some previously omitted experiments and added new results. This provides a clearer presentation of the contributions of our method.
>
> | #   | Single View Patch | Rendered Panoramic Patch | Instance | Zone | R2R-CE Val  Uneen   | R2R-CE Val  Uneen   | R2R-CE Val  Uneen  | R2R-CE Val   Uneen  | REVERIE-CE Val  Uneen | REVERIE-CE Val  Uneen | REVERIE-CE Val  Uneen | REVERIE-CE Val  Uneen |
> | --- | ----------------- | ------------------------ | -------- | ---- |:--------------:|:-------------:|:------------:|:-------------:|:--------------:|:--------------:|:--------------:|:--------------:|
> |     |                   |                          |          |      | NE$\downarrow$ | OSR$\uparrow$ | SR$\uparrow$ | SPL$\uparrow$ | NE$\downarrow$ | OSR$\uparrow$  | SR$\uparrow$   | SPL$\uparrow$  |
> | 1   | X                 | ✓                        | ✓        | ✓    | 5.34           | **62.1**      | **52.9**     | **45.7**      | **6.22**       | **48.9**       | **40.1**       | **28.5**       |
> | 2   | X                 | X                        | ✓        | ✓    | 7.59           | 43.9          | 32.7         | 21.1          | 8.76           | 27.4           | 19.3           | 11.4           |
> | 3   | X                 | ✓                        | X        | ✓    | 5.77           | 54.7          | 47.9         | 43.3          | 6.58           | 38.5           | 26.8           | 21.0           |
> | 4   | X                 | ✓                        | ✓        | X    | **5.26**       | 61.8          | 52.4         | 45.7          | 6.37           | 46.2           | 39.3           | 26.2           |
> | 5   | X                 | ✓                        | X        | X    | 5.63           | 51.1          | 45.7         | 40.2          | 7.89           | 34.8           | 25.7           | 17.8           |
> | 6   | ✓                 | X                        | ✓        | ✓    | 6.14           | 51.7          | 43.1         | 33.9          | 7.85           | 35.9           | 28.4           | 21.8           |
>
> **Table 1. More experiments for ablation study.**
>
>
> ### **1. The contribution of rendered panoramic 3D tokens, instance tokens, and zone tokens.**
>  As shown in Table 1, the removal of rendered panoramic patch tokens (Row 2), instance tokens (Row 3), zone tokens (Row 4), or both instance and zone tokens (Row 5) from the full model (Row 1) lead to performance drops of varying degrees. **The impact on performance follows the order: patch > instance > zone tokens**.
>
> ### **2. The significance of complex representation designs, such as rendering panoramic tokens using g3D-LF and obtaining instance-level representations via FastSAM. Their effectiveness for pre-exploration and lifelong memory.**
> As analyzed in VLN-3DFF[1], rendered panoramic patch tokens substantially mitigate the limited field of view caused by monocular cameras on most robots. **Comparing Rows 1 and 6 in Table 1, replacing rendered panoramic tokens with CLIP patch features extracted from the current monocular view** results in about a **10% drop** in navigation success rates on both R2R-CE and REVERIE-CE. Moreover, lifelong memory experiments in Table 3 of the original submission show that tokens rendered from feature fields have greater potential to leverage pre-exploration (although the comparison is not entirely fair)  and lifelong memory for performance improvement than the methods that directly extract patch-level features from current observations.
> For obtaining instance-level tokens via FastSAM, rows 1 and 3 in Table 1 demonstrate the performance gains brought by instance tokens. Moreover, instance-level representations are highly valuable for object-centric tasks in large-scale scenes when extending 3D-VLMs, such as instance grounding and mobile manipulation, although Dynam3D has not yet attempted such applications.
>
> ### **3. The heterogeneity of different types of tokens, such as patch, instance, and zone tokens, does not overly confuse the model learning.**
> Although we do not explicitly define learnable token-type embeddings, each token type has its own positional encoding network. This allows the 3D-VLM to distinguish between different token types. Additionally, during representation pretraining, patch, instance and zone tokens are aligned or distilled into the CLIP semantic space using corresponding language and 2D visual features at varying granularities. Despite sharing the same semantic space, each token type exhibits distinct semantic granularities.
>
> ### **4. Merging Discriminator Accuracy.**
> Evaluated on MP3D scenes from the R2R-CE Val Unseen split, the merging discriminator achieves **78.6%** accuracy in determining the merger of 2D and 3D instances. Although not perfect, the frustum culling strategy removes most near-field old instances, leaving only instances near the view boundaries and at a distance to be merged. Thus, the current accuracy is sufficient to effectively support instance token updates.
>
> ### **5. More details on the processing of VLN data.**
> As detailed in the Supplementary Material (Line 503 of the original submission), we use a pre-trained waypoint predictor[2] to generate candidate waypoints and select the GT waypoint closest to the ground-truth (GT) path and destination via the simulator API. The agent can achieve **96 SPL and nearly 100% success rate (SR)** on R2R-CE Val Unseen when it strictly **follows these GT waypoints**. This provides reliable supervision for training. Each next GT waypoint yields a heading angle (in 15° increments) and a forward distance (in 25 cm increments), which are used as GT textual action supervision for the 3D-VLM model.
>
> [1] Wang et al. Sim-to-real transfer via 3d feature fields for vision-and-language navigation. In 8th Annual Conference on Robot Learning, 2024.
>
> [2] Hong et al. Bridging the gap between learning discrete and continuous environments for vision-and-language navigation. In Proceedings of the IEEE/CVF Conference on Computer Vision and Pattern Recognition (CVPR), June 2022.

---

> > ### Comment · Reviewer_QXMr · 2025-08-05
> >
> > Thanks for the authors' response. My concerns are addressed. By the way, I noticed that the training costs more than 10 days overall given the large-amount data. So, my additional question is that given the large scale and costs, is it possible to train a comparable navigation policy with much less data? What data matters most?

---

> > > ### Author Response · Authors · 2025-08-05
> > > **Question about Training Cost**
> > >
> > > ### **Training Cost**
> > > Thank you very much for your reply. The long training time is mainly due to the limitations of our training resources. Specifically, we used:
> > >
> > > * 4 RTX 6000 Ada GPUs for 3D representation pre-training (8 days)
> > > * 2 RTX 6000 Ada GPUs for the Navigation model (9 days)
> > > * A total of approximately 1200 GPU hours
> > >
> > > Compared to other video-based VLA models, our training efficiency is not at a disadvantage:
> > >
> > > * **NaVILA:** 128 A100 GPUs for pre-training and 32 A100 GPUs for fine-tuning (~4000 GPU hours)
> > > * **NaVid:** 24 A100 GPUs (~672 GPU hours)
> > > * **Uni-NaVid:** 40 H800 GPUs (~1400 GPU hours)
> > >
> > > The substantial computational burden is primarily twofold: the immense parameter count of large models and the high number of predicted action steps, approximately 40 steps, required to complete a single navigation episode.
> > >
> > > A promising strategy, as outlined in our response to Reviewer kSWU under **Optimize inference latency with better action prediction policies**, involves reducing the frequency of large model inference. For instance, the large model could be used exclusively for high-level encoding and reasoning (e.g., predicting a distant sub-goal). A smaller action expert model could then leverage the large model's latent features to rapidly predict multi-step, low-level atomic actions. By reducing the number of the large model's run steps to 10+ per episode, training costs could be significantly mitigated.
> > >
> > >
> > > ### **Training Data**
> > > Compared to Video-based VLM methods, which transfer knowledge from large models pretrained on vast internet data with only minor architectural adjustments, Dynam3D employs different patch-instance-zone 3D representations. This substantial architectural change requires more extensive 3D-language data for training the 3D representation model and sufficient navigation data to ensure robust navigation generalization. Nevertheless, our experimental results highlight the superiority of our dynamic 3D representation-based 3D-VLM model for VLN tasks, presenting a promising alternative to video-based solutions.
> > >
> > > We believe there is still a way to train high-performance navigation policies with a small amount of data: **3D target grounding supervision**. A major reason for the difficulty in training current navigation models is the sparse alignment between navigation actions and the environment/instruction semantics. Introducing target grounding as part of the navigation model's output and training supervision could provide a stronger semantic alignment, significantly reduce data requirements, and accelerate convergence.

---

> > > > ### Comment · Reviewer_QXMr · 2025-08-05
> > > >
> > > > Thanks for your response. Be chill. My additional question is just for discussion, and I am not judging the efficiency of your method :) I know that with such a limited computing resource, it is quite hard to compete against prior works that use tremendous GPUs for training. So, my question is that, more specifically, do we really need that large amount of navigation data for training a strong navigation policy? And, if it is not, what is the most efficient data recipe (e.g., R2R, etc.)?

---

> > > > > ### Author Response · Authors · 2025-08-05
> > > > >
> > > > > Thank you so much for your reply and our engaging discussion!
> > > > >
> > > > > In fact, our past experience has shown that **data is crucial for the generalization performance of navigation models, with a very significant scale-up effect.** It is difficult to achieve strong generalization performance using only small-scale, manually annotated data like R2R, even if the model performs excellently on the R2R benchmark.
> > > > >
> > > > > Datasets like R2R only provide step-by-step instruction-following capabilities. In our real-world experiments or visitor demos, if the given instructions fall outside the dataset's domain, models trained solely on a small, single dataset like R2R perform very poorly. For example, if an instruction is a question ("Can you get me a glass of water?"), a rough description ("Please go to the living room and get me a glass of water"), or a demand-based statement ("Hi, robot, I'm a little thirsty"), the performance is quite bad.
> > > > >
> > > > > Therefore, in addition to improving metrics on standard VLN benchmarks, navigation training data that is **diverse in scenes (1K+ scenes), diverse in instructions (instruction-following R2R, RxR, ScaleVLN; inquiries NavRAG; coarse-grained descriptions REVERIE, CHORES; demand-based statements NavRAG), and massive in scale (1M+ navigation samples)** is still essential for real-world navigation performance.
> > > > >
> > > > > Certainly, strategies do exist for achieving strong generalization with limited navigation data, but these often depend on fully utilizing the prior knowledge of existing large models. This is precisely the reason why large video-based models try to minimize changes to their model architecture.
> > > > >
> > > > > However, for navigation tasks involving large-scale 3D environment understanding, 2D or video-based VLMs trained on internet data will always face a domain gap and have limitations in 3D comprehension. Therefore, we believe that VLMs based on 3D or even 4D (3D + time) representations still have a higher performance ceiling.

---

> ### Comment · Reviewer_QXMr · 2025-08-05
>
> Thanks for your sharing. Good work.

---

### Official Review · Reviewer_V8c4 · 2025-07-03

**Clarity:** 2
**Significance:** 3
**Originality:** 3
**Rating:** 5
**Confidence:** 3

**Summary:**

This paper presents Dynam3D, a dynamic layered 3D representation framework for vision-and-language navigation that addresses limitations of existing video-based approaches through three primary contributions. First, the authors develop a hierarchical patch-instance-zone representation model that projects 2D CLIP features into 3D space and maintains multi-view consistent 3D instances using a merging discriminator trained on large-scale 3D segmentation data. Second, they implement a dynamic update mechanism employing frustum culling to remove outdated features and real-time hierarchical updates to adapt to changing environments. Third, they pass in these representations into a 3D Vision-Language Model that combines panoramic 3D patch tokens.

**Questions:**

N/A - see weaknesses

**Ethical Concerns:**

["NO or VERY MINOR ethics concerns only"]

**Final Justification:**

All followup questions have been answered, nothing remains unresolved. I would like to keep my initial high score.

**Quality:**

3

**Strengths And Weaknesses:**

Strengths:
* Technical innovation: Clearly identifies gaps in the current SOTA and addresses it with their method.
* Results: Demonstrates clear strength in performance in both simulated and real world setting.

Weaknesses:
* It is unclear how this method scales to larger environments - especially outdoors.
* By extension, it is unclear to me how this method would work if frames end up being drastically different from each other.

---

> ### Author Rebuttal · Authors · 2025-07-30
>
> ## **1. Handling Large-Scale Scenes, e.g., Outdoors**
> Thanks for your valuable comments! Although currently primarily designed for indoor navigation, our Dynam3D model architecture is in fact also suitable for outdoor navigation.
> ### **1) Representation and Computational Overhead**
> Unlike fine-grained voxel-based methods, our instance and zone-level tokens effectively handle large-scale scene representation without introducing an excessive number of tokens and huge computational overhead. The number of 3D patch tokens rendered via feature fields is fixed at 12x48. Furthermore, the KD-Tree feature retrieval and sparse sampling strategies employed in HNR and g3D-LF ensure highly efficient feature field rendering. Even in outdoor urban scenes, a single RTX 4090 GPU can render 12x48 panoramic patch tokens at over 15 FPS.
> ### **2) Generalization and Training Data**
> Although the model architecture of our Dynam3D can be adapted to large scenes and outdoor navigation, its current generalization capabilities for outdoor navigation are limited because the training data of both the 3D representation and the 3D-VLM model originate from indoor environments. However, it’s possible to fine-tune and adapt the model using outdoor navigation datasets such as Touchdown, map2seq, and VLN-VIDEO.
> ### **3) Outdoor Robot Deployment**
> For indoor navigation, the Stretch 3 robot used in the Dynam3D experiments is equipped with a precise SLAM system, accurate camera pose estimation, and sufficient depth camera precision. However, the localization system and depth camera do not perform well in outdoor environments due to the narrow field-of-view of the LiDAR on the robot. Nonetheless, this issue can be easily mitigated by equipping the robot with a wide-angle 4D-LiDAR to increase the field-of-view. Our Dynam3D model supports sensory inputs from the wide-angle LIDAR with no need for drastic modifications.
>
> ## **2. Handling Large Inter-Frame Changes**
> The frustum culling of Dynam3D and its token addition, deletion, and update strategies are well-suited to handle significant changes between observation frames.
> ### **1)  Large robot and camera movements leading to significant inter-frame changes.**
> In such scenarios, the overlap between the current frame and historical frames is minimal.  Our Dynam3D handles this situation effectively by primarily adding new patch, instance, and zone tokens. For example, it directly encodes 2D instances as new 3D instances as described in lines 152–156 of the original paper. This eliminates the need for extensive 2D-to-3D instance merging.
> ### **2) Rapid movement of people or objects within the scene causing huge inter-frame changes.**
> Frustum culling can effectively delete old features and update new ones when a moving object moves away from or traverses out of the camera field-of-view. However, frustum culling may fail to remove outdated features from earlier frames when objects move closer to the camera. This is due to occlusions caused by objects in the new frame. A potential future improvement is to address this issue using 3D object or point tracking methods.

---

### Decision · Program_Chairs · 2025-09-17

**Decision:**

Accept (oral)

**Comment:**

Summary:
This paper presents Dynam3D, a dynamic layered 3D representation framework for vision-and-language navigation. It integrates patch, instance, and zone tokens with hierarchical updates, enabling robust spatial understanding, long-term memory, and adaptability to dynamic environments. The method achieves state-of-the-art results across multiple benchmarks and demonstrates strong real-world robot performance.

Strengths:
The technical design is solid and innovative: it brings hierarchical 3D awareness into VLN with carefully engineered dynamic updates. Experiments are extensive, covering both simulation and real-world settings, and show clear gains over baselines. The work is impactful and addresses practical deployment.

Weaknesses:
The pipeline is somewhat complex and training is resource-intensive, and questions remain about scalability outdoors and efficiency. However, the authors’ rebuttal addressed these concerns with convincing clarifications and ablations.

Overall, this is a technically strong and comprehensive contribution with clear empirical value. ACs recommend acceptance.